# Evaluation of risk adjustment performance of diagnosis-based and medication-based comorbidity indices in patients with chronic obstructive pulmonary disease

**Huei Guo Ie[1]⊛, Chao-Hsiun Tang[2]⊛, Mei-Ling Sheu[2]⊛, Hung-Yi Liu[3], Ning Lu[4], Tuan-Ya Tsai[5], Bi-Li Chen[6], Kuo-Cherh Huang**[2]*

**1** Teaching Resource Center, Office of Academic Affairs, Taipei Medical University, Taipei City, Taiwan, **2** School of Health Care Administration, College of Management, Taipei Medical University, Taipei City, Taiwan, **3** Health and Clinical Research Data Center, Taipei Medical University, Taipei City, Taiwan, **4** Department of Health Administration, College of Health and Human Services, Governors State University, University Park, Illinois, United States of America, **5** Department of Pharmacy, Taipei Medical University-Shuang Ho Hospital, New Taipei City, Taiwan, **6** Department of Pharmacy, Taipei Medical University Hospital, Taipei City, Taiwan

⊛ These authors contributed equally to this work.
* kchuang@tmu.edu.tw

**Data Availability Statement:** The data underlying this study are from the National Health Insurance Research Database (NHIRD) which has been

## Abstract

### Objectives

This study assessed risk adjustment performance of six comorbidity indices in two categories of comorbidity measures: diagnosis-based comorbidity indices and medication-based ones in patients with chronic obstructive pulmonary disease (COPD).

### Methods

This was a population–based retrospective cohort study. Data used in this study were sourced from the Taiwan National Health Insurance Research Database. The study population comprised all patients who were hospitalized due to COPD for the first time in the target year of 2012. Each qualified patient was individually followed for one year starting from the index date to assess two outcomes of interest, medical expenditures within one year after discharge and in-hospital mortality of patients. To assess how well the added comorbidity measures would improve the fitted model, we calculated the log-likelihood ratio statistic $G^2$. Subsequently, we compared risk adjustment performance of the comorbidity indices by using the Harrell $c$-statistic measure derived from multiple logistic regression models.

### Results

Analytical results demonstrated that that comorbidity measures were significant predictors of medical expenditures and mortality of COPD patients. Specifically, in the category of diagnosis-based comorbidity indices the Elixhauser index was superior to other indices, while the RxRisk-V index was a stronger predictor in the framework of medication-based

transferred to the Health and Welfare Data Science Center (HWDC). The Taiwan government prohibits release of the aforementioned datasets to the public domain. Interested researchers can obtain the data through formal application to the HWDC, Department of Statistics, Ministry of Health and Welfare, Taiwan (https://dep.mohw.gov.tw/DOS/np-2497-113.html).

**Funding:** this study was supported by the Ministry of Science and Technology in Taiwan (grant number MOST 105-2410-H-038-010). The funder had no role in study design, data collection and analysis, decision to publish, or preparation of the manuscript.

**Competing interests:** The authors declared no potential conflicts of interest with respect to the research, authorship, and/or publication of this article.

codes, for gauging both medical expenditures and in-hospital mortality by utilizing information from the index hospitalization only as well as the index and prior hospitalizations.

## Conclusions

In conclusion, this work has ascertained that comorbidity indices are significant predictors of medical expenditures and mortality of COPD patients. Based on the study findings, we propose that when designing the payment schemes for patients with chronic diseases, the health authority should make adjustments in accordance with the burden of health care caused by comorbid conditions.

## Introduction

Health care outcome measures, such as mortality and healthcare resource utilization, require effective risk adjustment based on patient characteristics and comorbidities that exist prior to the episode of care [1–4]. Population-based healthcare administrative databases are increasingly used by researchers and policymakers since they are relatively inexpensive and invaluable data resources for effectiveness and outcomes research, health economics analysis, health services research, and evidence-informed health policy-making. To measure comorbidities, numerous indices have been developed and validated with utilizations of healthcare administrative databases. Yet there are inherent limitations of such databases since they are originally gathered for administrative or billing purposes other than academic research. The main advantages of comorbidity indices derived from administrative healthcare databases are their real-life setting, relatively low cost of data acquisition, and time efficiency of capturing comorbid conditions of entire populations or disease cohorts with long follow-up duration. Nonetheless, there are inherent limitations that affect both the completeness and validity of administrative healthcare data, including the lack of important prognostic indicators and lifestyle information as well as the accuracy of diagnostic and procedural codes, which can introduce bias when investigators aim to assess health services utilization, medical expenditures, and quality of care of patients [5–7]. Consequently, when designing and interpreting the results of studies that rely on information extracted from nonclinical databases, researchers recognize that great care must be taken.

For all research pertaining to health-related outcome measures (such as mortality, hospitalization, and medical expenditures), chief among the challenges of converting claims data into research-appropriate analytic files is to adequately risk-adjust for comorbidities as to get unbiased estimates [8–10]. A comorbidity means a pre-existing health condition that coexists with an index disease and may impact on treatment outcomes such as increased mortality, decreased quality of life, and increased utilization of healthcare services compared to patients with no comorbidity [11–13]. Along with the increasing use of administrative healthcare databases, a number of claims-based comorbidity measures have been constructed as proxy measures of overall health status of patients.

In the health services literature, widely used risk adjustment models based on coded comorbidities include the Charlson comorbidity index (CCI) [14], the Charlson/Deyo index [15], the Charlson/D'Hoore index [16], the Charlson/Romano index [17], and the Elixhauser index (EI) [18]. Those indices are based on a standard system for coding diagnoses, the International Classification of Diseases, Ninth Revision, Clinical Modification (ICD-9-CM) codes, from administrative health data of hospitalization or outpatient visit. Another distinctive class of

comorbidity instruments includes those indices using medication dispensing data to measure the burden of comorbid conditions; for example, the chronic disease score (CDS) [19], the modified chronic disease score (CDS-2) [20], the RxRisk index [21], and the RxRisk-V index [22]. Hence, the aforementioned comorbidity indices were included in this analysis, and described at greater length in the Methods section.

Chronic obstructive pulmonary disease (COPD) is characterized by persistent airflow obstruction that is usually progressive and only partly reversible [23]. COPD is recognized as an important global public health challenge because it is increasing in prevalence and becomes a major and growing source of morbidity and mortality in countries at all levels of economic development [24, 25]. There has been a growing recognition that comorbidities (such as cardiovascular disease) are likely to be present in a greater proportion of patients with COPD compared to the general population, and reportedly have a negative effect on prognosis and survival of those patients [26, 27].

Perhaps due to the fact that there is no gold standard measure of comorbidity, the literature offers no clear consensus on risk adjustment performance of various claims-based comorbidity measures. With this in mind, the objective of this population-based retrospective cohort study was to compare the performance of predicting medical expenditures and mortality in patients with COPD among various comorbidity indices. Specifically, four diagnosis-based comorbidity indices (the Deyo index, the Romano index, the D'Hoore index, and the Elixhauser index) and two medication-based comorbidity indices (the modified chronic disease score [the CDS-2] and the RxRisk-V index) were evaluated since they have been used frequently in the health services literature to adjust for baseline health status along with the consideration of the feasibility of extracting comorbidity index information from the database of this study.

## Methods

### Data sources and the study population

Data used in this study were mainly sourced by unique national identification numbers of the study population from the Taiwan National Health Insurance Research Database (NHIRD) which encompasses insurance claims from over 99% of the population of Taiwan of more than 23 million people and is currently maintained by the Health and Welfare Data Science Center, Ministry of Health and Welfare, Taiwan. Taiwan launched a universal single-payer National Health Insurance programme in 1995. The NHIRD provides comprehensive information on health care resource utilization rendered in the inpatient and outpatient settings, including diagnosis codes, procedure claims, and medication records. In the literature, validation of standard ICD codes and algorithms has been established [28–30]. In a similar vein, in the context of Taiwan's health care services and coding practices the NHIRD has been demonstrated to have high validity [31, 32].

Data in the NHIRD that could be used to identify patients or care providers, including medical institutions and physicians, are scrambled cryptographically and then released in electronic format to the public annually for research purposes by the National Health Research Institute of Taiwan. Since the present study utilized de-identified secondary data, it was exempt from full review by the Institutional Review Board of Taipei Medical University, Taiwan (TMU-JIRB No. N201605057). The need for participant consent was waived by the Institutional Review Board.

The study population comprised all patients who were hospitalized due to chronic obstructive pulmonary disease (COPD; ICD-9-CM codes: 491.x, 492.x, 496.x) for the first time in the target year of 2006, 2009, or 2012. The codes and algorithms had been validated and found to have a sensitivity of 85.0% and a specificity of 78.4% [33]. Each qualified patient was

individually followed for one year starting from the index date to compare the discriminatory power of various comorbidity indices as regards two outcomes of interest, medical expenditures within one year after discharge and in-hospital mortality of patients. The data period used in this analysis was from year 2005 to year 2013, whereas year 2005 was selected to assess the eligibility of sample patients with the index dates in year 2006, and year 2013 was used to retrieve data of outcome measures of patients entering the study cohort in year 2012 (i.e., a 1-year lookback period). We further assessed the robustness of our models by repeating the analyses with three different target years; years 2006, 2009, and 2012.

## Comorbidity indices

This study encompassed two categories of comorbidity measures: diagnosis-based comorbidity indices (the CCI, the Charlson/Deyo, the Charlson/D'Hoore, the Charlson/Romano, and the Elixhauser index) and medication-based ones (the CDS, the RxRisk, and the RxRisk-V index), as detailed below. The CCI index was created by Charlson and colleagues [14] by using chart review to predict 1-year mortality in a cohort of 604 hospitalized patients in 1984. The index was revised in 1987 by including a list of 19 comorbid conditions, with each condition assigned a weight of 1, 2, 3, or 6, based on adjusted hazard ratios for each condition derived from Cox proportional hazards regression models. All of the individual weights were then added up to create a single comorbidity score for each patient. As for the Charlson/Deyo index, Deyo et al. [15] amended the CCI by identifying the ICD-9-CM diagnosis and procedure codes corresponding to each of the 19 comorbid conditions proposed by Charlson and colleagues. The codes for leukemia and lymphoma were combined in the "any malignancy" category, and thus there was a list of 17 comorbid conditions for the Deyo CCI. As regards the Charlson/D'Hoore index [16], D'Hoore et al. adapted the CCI by using only the first three digits of ICD-9 coding without CM (since it is the coding fashion of many healthcare institutions outside the US). In addition, due to the likelihood that coding of the tailing digits in ICD-9 codes may lead to inconsistencies, therefore, D'Hoore et al. had declared that the Charlson/ D'Hoore index was a more reliable comorbidity measure. The Charlson/Romano index [17], originally termed as the Dartmouth-Manitoba CCI, was firstly created by Roos et al. in 1989 and subsequently modified by Romano and colleagues in 1993. Compared with the Deyo CCI, the Romano CCI contains more ICD-9-CM codes. Concerning the Elixhauser index [18] was developed by Elixhauser and colleagues with a list of 30 comorbidities. In the literature there is strong evidence that the Elixhauser index outperforms the CCI, but the CCI continues to be widely used. One disadvantage of the EI is that unlike the CCI which produces a single comorbidity score on a continuous scale for each patient, the Elixhauser index entails 30 dichotomous variables but no weighting system to create a single score, making its use for analysis of comorbidity burdensome.

With respect to the set of medication-based comorbidity indices, the CDS, the first pharmacy-based measure of comorbidity, was created by von Korff and colleagues [19] in 1992. The methodology was based on medications rather than diagnostic codes to identify comorbid conditions of patients. A panel of experts was convened to evaluate patterns of utilization of selected medications as to create comorbidity categories, and weights were apportioned by consensus. The CDS consists of 17 comorbidity categories. Clark and colleagues [20] subsequently updated and modified the original CDS by expanding the disease categories to 28 as well as updating medications, and also assigned a weight to each disease category based on results of regression models. With reference to the RxRisk index [21], it includes 57 disease categories and associated medication classes, and was originally developed as a risk assessment instrument by using outpatient pharmacy data to ascertain chronic diseases. As for the

RxRisk-V index [22], it is a subsequent modification of the RxRisk-V index, consisting of 45 categories of comorbidity adapted to the United States Veterans Health Administration population.

## Statistical analysis

We intended to assess if adding comorbidity measures to the baseline model would significantly improve the predictive capacity of the model, whereas the baseline model (containing no comorbidity information) included age and gender of the patient, if surgery undertaken when hospitalized, and the length of hospital stay. Firstly, medical expenditure data revealed a positively skewed distribution, and thus they were converted to natural logarithm values. For all logistic regression models, medical expenditures were then dichotomized and the threshold was set at Q3 (the 75th percentile), with Q1-Q3 in the low-cost group whereas Q4 in the high-cost group, in accordance with previous research [34, 35]. Furthermore, to assess how well the added comorbidity measures would improve the fitted models, we firstly calculated the log-likelihood ratio statistic $G^2$ [36]. Subsequently, we measured and compared risk adjustment performance of various comorbidity indices by using the Harrell c-statistic measure (c-statistic) derived from multiple logistic regression models [37]. The c-statistic is a measure of concordance between model-based risk estimates and observed events, and thus provides an assessment of the performance of a predictive model. The c-statistic ranges from zero to one, with a value below 0.5 indicating a very poor model, 0.5 representing chance prediction, while 1.0 demonstrating perfect prediction. In general, a c-statistic of 0.7 indicates adequate prediction, 0.8 is very good, and 0.9 or more represents excellent predicting capabilities [38].

Furthermore, there were two data periods used in this analysis. The first data period was the index hospitalization, and the other one was the index and prior 1-year hospitalizations. To put it another way, the index and prior 1-year hospitalizations contained a 1-year lookback period, while the index hospitalization didn't have a lookback period. The index hospitalization was identified as the first hospitalization of a sample patient during the three target years of 2006, 2009, and 2012, respectively.

To characterize the power of this study, we employed the PROC POWER statement in the POWER procedure of the SAS/STAT software (SAS Institute, Cary, NC, USA). The computed study power was 0.824.

All analyses were performed using the SAS software version 9.4 (SAS Institute, Cary, NC, USA). Statistical significance was set at $P < 0.05$ (two-tailed).

## Results

For the three target years (2006, 2009, and 2012), there were 3,367, 3,191, and 3,220 COPD patients met our inclusion criteria, respectively. The ratio of male to female patients was roughly 1.8:1, and the mean age was about 69.5. One-year mean medical costs were New Taiwan Dollar (NT$) 167,015.2 (year 2006), NT$154,198.6 (2009), and NT$129,605.4 (2012) (average exchange rate from year 2006 to year 2012: 1 U.S. Dollar = NT$31.45). The in-hospital mortality rates among study samples were 0.30% (year 2006), 0.41% (2009), and 0.34% (2012). Demographic and clinical profiles of those selected patients were presented in Table 1.

Table 2 presents the log-likelihood ratio statistic $G^2$ indicating the extent of how well the added comorbidity measures would improve the nested baseline model (included patient's age and gender, if surgery undertaken when hospitalized, and the length of hospital stay) in terms of medical expenditures and in-hospital mortality. Among the six comorbidity indices, the RxRisk-V index outperformed others concerning the improvement of the fit of the regression models (based on $G^2$ values) for both medical expenditures and in-hospital mortality in all

**Table 1. Demographic and clinical characteristics of the study population of the selected data periods.**

| Variables | | 2006 (n = 3,367) | | 2009 (n = 3,191) | | 2012 (n = 3,220) | |
|---|---|---|---|---|---|---|---|
| Gender | | | | | | | |
| | Male | 2,117 | (62.87%) | 2,029 | (63.59%) | 2,143 | (66.55%) |
| | Female | 1,250 | (37.13%) | 1,162 | (36.41%) | 1,077 | (33.45%) |
| Age in years (mean ± SD[a]) | | 68.7 ± 13.12 | | 69.8 ± 13.15 | | 70.0 ± 13.03 | |
| If undergoing surgery | | | | | | | |
| | Yes | 287 | (8.52%) | 240 | (7.52%) | 250 | (7.76%) |
| | No | 3,080 | (91.48%) | 2,951 | (92.48%) | 2,970 | (92.24%) |
| If being hospitalized | | | | | | | |
| | Yes | 619 | (18.38%) | 537 | (16.83%) | 538 | (16.71%) |
| | LOS[b] | 10.1 ± 12.48 | | 10.4 ± 14.75 | | 9.8 ± 10.62 | |
| | No | 2,748 | (81.62%) | 2,654 | (83.17%) | 2,682 | (83.29%) |
| One-year medical costs (NT$[c]) (mean ± SD[a]) | | 167,015.2 ± 394,409.45 | | 154,198.6 ± 297,575.52 | | 129,605.4 ± 247,159.34 | |
| | Q3[d] | | 154,028 | | 153,882 | | 129,908 |
| | logQ3[e] | | 11.94 | | 11.94 | | 11.77 |
| In-hospital mortality | | | | | | | |
| | Yes | 10 | (0.30%) | 13 | (0.41%) | 11 | (0.34%) |
| | No | 3,357 | (99.70%) | 3,178 | (99.59%) | 3,209 | (99.66%) |

[a]SD, standard deviation.

[b]LOS, length of stay.

[c]NT$, New Taiwan Dollar. Average exchange rate from 2006 to 2012: 1 U.S. Dollar = NT$31.45.

[d]Q3, the third quartile.

[e]logQ3, the natural logarithm of Q3.

three target years, followed by the model with the Elixhauser score. The same comparative results could be observed for both strategies of using the index hospitalization only and the index and prior hospitalization.

Results of different models indicating risk adjustment performance of various comorbidity indices in predicting medical expenditures and mortality are presented in Table 3. Overall, *c*-statistics for those multiple logistic regression model specifications ranged from 0.687 to 0.851. The model with the Elixhauser index was comprehensively a better comorbidity risk adjustment with higher *c*-statistics relative to other indices. Specifically, the Elixhauser index added higher predicting capabilities when using the index hospitalization only as well as index and prior hospitalizations, compared with other comorbidity methods, in both year 2006 and year 2012. The highest *c*-statistic was 0.851 for the Elixhauser index when using information from the index hospitalization only in predicting in-hospital mortality of year 2012.

## Discussion

Adjustment for comorbidity is critical in observational studies because baseline differences in health status between study groups may modulate differences detected in research outcomes. Hence, this investigation appraised and compared risk adjustment performance of two categories of comorbidity measures: diagnosis-based comorbidity indices (the Deyo index, the Romano index, the D'Hoore index, and the Elixhauser index) and medication-based ones (the CDS-2 and the RxRisk-V index). Although some work has been done to compare diagnosis- and medication-based comorbidity indices (e.g., the Cortaredona study [39] in 2017), more research from different population and datasets is justified as to establish the respective merits

**Table 2. $G^2$ statistics of different models indicating the contributions of various comorbidity indices to the baseline model.**

| | 2006 | | | | 2009 | | | | 2012 | | | |
|---|---|---|---|---|---|---|---|---|---|---|---|---|
| | Medical expenditures | | In-hospital mortality | | Medical expenditures | | In-hospital mortality | | Medical expenditures | | In-hospital mortality | |
| | $G^2$ | $p$ | $G^2$ | $p$ | $G^2$ | $p$ | $G^2$ | $p$ | $G^2$ | $p$ | $G^2$ | $p$ |
| *Index hospitalization only* | | | | | | | | | | | | |
| Baseline model + Deyo | 27.71 | 0.006 | 18.48 | 0.747 | 32.98 | 0.001 | 15.95 | 0.194 | 24.26 | 0.012 | 16.43 | 0.843 |
| Baseline model + D'Hoore | 36.18 | 0.002 | 17.94 | 0.266 | 36.42 | < 0.001 | 14.39 | 0.421 | 32.37 | 0.004 | 15.44 | 0.346 |
| Baseline model + Elixhauser | 49.43 | 0.004 | 35.91 | 0.042 | 38.14 | < 0.001 | 37.44 | 0.029 | 38.98 | 0.002 | 58.59 | < 0.001 |
| Baseline model + Romano | 29.53 | 0.003 | 18.86 | 0.716 | 36.35 | 0.006 | 15.40 | 0.221 | 29.15 | 0.024 | 16.34 | 0.176 |
| Baseline model + Revised CDS[a] | 30.67 | 0.032 | 27.63 | 0.377 | 36.61 | 0.005 | 26.02 | 0.352 | 24.69 | 0.012 | 21.65 | 0.542 |
| Baseline model + RxRisk-V | 51.52 | 0.028 | 62.90 | 0.007 | 72.79 | < 0.001 | 55.69 | 0.011 | 41.92 | 0.036 | 71.05 | < 0.001 |
| *Index and prior hospitalizations* | | | | | | | | | | | | |
| Baseline model + Deyo | 38.55 | 0.001 | 15.86 | 0.391 | 42.03 | < 0.001 | 22.32 | 0.100 | 52.68 | < 0.001 | 19.11 | 0.172 |
| Baseline model + D'Hoore | 30.78 | 0.009 | 21.73 | 0.703 | 48.55 | < 0.001 | 31.63 | 0.169 | 40.35 | < 0.001 | 27.32 | 0.340 |
| Baseline model + Elixhauser | 60.82 | < 0.001 | 37.53 | 0.011 | 56.59 | < 0.001 | 44.62 | 0.009 | 64.21 | < 0.001 | 43.24 | 0.018 |
| Baseline model + Romano | 40.46 | < 0.001 | 13.39 | 0.572 | 42.71 | < 0.001 | 23.45 | 0.075 | 55.28 | < 0.001 | 23.25 | 0.083 |
| Baseline model + Revised CDS[a] | 60.27 | < 0.001 | 26.99 | 0.211 | 47.08 | 0.005 | 26.07 | 0.404 | 52.80 | 0.001 | 25.07 | 0.458 |
| Baseline model + RxRisk-V | 68.85 | < 0.001 | 62.90 | 0.007 | 71.57 | < 0.001 | 71.05 | < 0.001 | 66.19 | 0.002 | 71.11 | < 0.001 |

[a]CDS, chronic disease score.

of each comorbidity index and the generalizability of comparative performance. Viewed in this way, this study adds real-world evidence from population-based datasets and different data periods to the body of knowledge about the utility of various comorbidity indices. In particular, we have evaluated the relative performance of four diagnosis-based and two medication-based comorbidity indices with regard to two outcome measures of medical expenditures and in-hospital mortality of COPD patient altogether.

Overall, this analysis demonstrated that comorbidity measures were significant predictors of medical expenditures and mortality of COPD patients. Specifically, in the category of diagnosis-based comorbidity indices the Elixhauser index was superior to other indices, while the RxRisk-V index was a stronger predictor in the framework of medication-based codes, for gauging both medical expenditures and in-hospital mortality by utilizing information from the index hospitalization only as well as the index and prior hospitalizations.

Much ink has been spilled on the predicting performance of various comorbidity measures either in specific populations (for example, patients with chronic obstructive pulmonary disease, human immunodeficiency virus infection, or cancer) [40–43], or for specific outcome measures (such as surgical outcomes, mortality, hospitalization, or medical expenditures) [9,

**Table 3. _C_ statistics of different models indicating the discriminatory power of various comorbidity indices predicting medical expenditures and mortality.**

| | 2006 | | | | 2009 | | | | 2012 | | | |
| | Medical expenditures | | In-hospital mortality | | Medical expenditures | | In-hospital mortality | | Medical expenditures | | In-hospital mortality | |
| | _c_ | $\Delta c$ (%)[b] | _c_ | $\Delta c$ (%)[b] | _c_ | $\Delta c$ (%)[b] | _c_ | $\Delta c$ (%)[b] | _c_ | $\Delta c$ (%)[b] | _c_ | $\Delta c$ (%)[b] |
|---|---|---|---|---|---|---|---|---|---|---|---|---|
| Baseline model[a] | 0.709 | | 0.739 | | 0.685 | | 0.733 | | 0.730 | | 0.815 | |
| _Index hospitalization only_ | | | | | | | | | | | | |
| Baseline model + Deyo | 0.723 | 1.97 | 0.759 | 2.71 | 0.691 | 0.88 | 0.735 | 0.27 | 0.739 | 1.23 | 0.835 | 2.15 |
| Baseline model + D'Hoore | 0.721 | 1.69 | 0.762 | 3.11 | 0.692 | 1.02 | 0.736 | 0.41 | 0.746 | 2.19 | 0.815 | 0.01 |
| Baseline model + Elixhauser | 0.733 | 3.39 | 0.833 | 12.72 | 0.708 | 3.36 | 0.754 | 2.86 | 0.748 | 2.47 | 0.851 | 2.42 |
| Baseline model + Romano | 0.720 | 1.55 | 0.752 | 1.76 | 0.698 | 1.90 | 0.734 | 0.14 | 0.743 | 1.78 | 0.817 | 0.25 |
| Baseline model + Revised CDS[c] | 0.711 | 0.28 | 0.768 | 3.92 | 0.702 | 2.48 | 0.729 | -0.55 | 0.733 | 0.41 | 0.819 | 0.49 |
| Baseline model + RxRisk-V | 0.714 | 0.71 | 0.766 | 3.65 | 0.687 | 0.29 | 0.748 | 2.05 | 0.736 | 0.82 | 0.813 | -0.25 |
| _Index and prior hospitalizations_ | | | | | | | | | | | | |
| Baseline model + Deyo | 0.731 | 3.10 | 0.741 | 0.27 | 0.713 | 4.09 | 0.735 | 0.27 | 0.764 | 4.66 | 0.818 | 0.37 |
| Baseline model + D'Hoore | 0.725 | 2.26 | 0.785 | 6.22 | 0.708 | 3.36 | 0.735 | 0.27 | 0.767 | 5.07 | 0.822 | 0.86 |
| Baseline model + Elixhauser | 0.735 | 3.67 | 0.814 | 1.15 | 0.725 | 5.84 | 0.752 | 2.59 | 0.769 | 5.34 | 0.823 | 0.98 |
| Baseline model + Romano | 0.728 | 2.68 | 0.765 | 3.52 | 0.722 | 5.40 | 0.732 | -0.14 | 0.768 | 5.12 | 0.817 | 0.25 |
| Baseline model + Revised CDS[c] | 0.727 | 2.54 | 0.772 | 4.47 | 0.721 | 5.26 | 0.741 | 1.09 | 0.759 | 3.97 | 0.821 | 0.74 |
| Baseline model + RxRisk-V | 0.730 | 2.96 | 0.793 | 7.31 | 0.714 | 4.23 | 0.737 | 0.55 | 0.761 | 4.25 | 0.815 | 0.01 |

[a]Variables in the baseline model included gender, age, if undergoing surgery, and length of stay.

[b]$\Delta c(\%) = [(c$ statistic of the specific model$-c$ statistic of the baseline model$)/c$ statistic of the baseline model$] \times 100\%$.

[c]CDS, chronic disease score.

44–50]. Review of the literature reveals that although the preponderance of studies indicate that the CCI is the most widely used claims-based comorbidity measure, growing evidence supports the notion that the Elixhauser index exhibits superior risk adjustment performance [7, 39, 42, 46, 48]. Furthermore, Schneeweiss and colleagues [44] reported that diagnosis-based comorbidity coding algorithms (such as the Romano index) generally performed better at predicting 1-year mortality than medication-based comorbidity indices (for example, CDS). Conversely, they also demonstrated that the number of distinct medications prescribed during a 1-year baseline period was the best predictor of future physician visits and medical expenditures. In their systematic review paper pertaining to comorbidity indices, Yurkovich and colleagues [5] suggested that a diagnosis-based index (e.g., the Romano index) be adopted in studies where the outcome measure was mortality, whereas a medication-based measure (such as the RxRisk-V index) be utilized for research when predicting health care utilization outcomes.

COPD is among the most prevalent chronic diseases and represent a heavy financial burden on healthcare systems worldwide [51]. A couple of implications in the field of comorbidity measures can be drawn from this investigation of the COPD study population. Firstly, this study demonstrated that all comorbidity indices assessed in this investigation were significantly predictors of medical expenditures of COPD patients, but exhibited moderate results only in regard to in-hospital mortality (Table 2). Among the six comorbidity indices, the medication-based RxRisk-V index provided the best fit relating to the improvement of the regression models for both medical expenditures and in-hospital mortality, followed by the diagnosis-based Elixhauser index (Table 2).

Furthermore, the overall results of this analysis suggested that the Elixhauser index exhibited the best risk adjustment performance in predicting both medical expenditures and mortality (Table 3). Findings of this study mostly confirm results from previous research as regards patients with COPD [40, 42]. For instance, in the Austin study [42] results revealed that the Elixhauser index ($c$-statistic = 0.822) exhibited slightly better risk adjustment performance than the Charlson index ($c$-statistic = 0.819) concerning predicting 1-year mortality in patients with COPD, while a medication-based index (the Johns Hopkins ADGs) had marginally higher predicting ability ($c$-statistic = 0.830) than both the Elixhauser and the Charlson indices. Those arguments are largely comparable to the present study's findings in respect of the similar study population.

The widely-used Charlson coding algorithm has been adopted for risk adjustment in studies of patients with major severe diseases in the literature [40–43, 48, 52]. However, even with the popularity of the CCI, previous studies have compared the relative performance of the Charlson and Elixhauser indices and have mostly reached the conclusion of the Elixhauser index outperforming the CCI [41–43, 48, 53–55]. For example, the Dominick study [53] established the superiority of the RxRisk-V and Elixhauser indices over the CCI in predicting health service use in patients with osteoarthritis. Similarly, Lieffers and colleagues [43] demonstrated that the Elixhauser comorbidity measure outperformed the CCI for colorectal cancer survival prediction. The Menendze study [54] also reached a similar conclusion as regards superior risk adjustment performance of the Elixhauser coding algorithm in predicting in-hospital mortality after orthopaedic surgery. Moreover, in their research focusing on patients with COPD, the same study population as the current investigation, Buhr and colleagues [56] concluded that the Elixhauser comorbidity index performed slightly better than the CCI in predicting the 30-day readmission risk. The aforementioned published findings are mostly in line with the results of this study.

It is worth noting that the RxRisk-V index is based on prescription medication use, whereas the Elixhauser measure is based on ICD-9-CM codes. Consequently, those comorbidity risk adjustment methods provide diverse options and feasibility for different health care institutions, depending on the types of medical administrative databases available [53]. In addition, it should be noted that evidence concerning the comparative performance of diagnosis-based and medication-based comorbidity indices is inconsistent. There are studies showing that the Romano index (diagnosis-based measure) has better predictive performance than the CDS (medication-based measure) in predicting 1-year mortality [10, 43]. Conversely, other research has revealed the reverse results, particularly when predicting medical expenditures and health service utilizations [44, 45, 57]. For instance, in the Perkins study [43] results demonstrated that medication-based comorbidity indices performed better than the CCI or the total number of chronic conditions in predicting total health care costs and the number of outpatient visits over one year. Farley and colleagues [58] gathered at the same conclusion by establishing that the prescription claims-based RxRisk-V index outperformed the Charlson and the Elixhauser diagnosis-based comorbidity indices in predicting healthcare expenditures.

The main strength of the study is that we take advantage of a nationwide, population-based registry database (the NHIRD), which renders our results more robust because potential validity threats of selection bias, recall bias, and information bias, inherited in cross-sectional or regional studies, would be minimalized. In addition, this analysis appraised risk adjustment performance of two categories of comorbidity measures: diagnosis-based comorbidity indices and medication-based ones. Prior studies indicated that combining several sources of morbidity information, such as hospital discharge information, diagnosis-based comorbidity measures, and pharmacy claims, could reduce residual confounding bias [10, 44]. Moreover, medication-based data are documented as a more complete, reliable, and timely data source than diagnosis-based data [22]. Nonetheless, data on prescription medication use are not readily available for the entire population in many jurisdictions. For example, in Ontario, Canada, data regarding prescribed medications are only available for seniors (those aged 65 and over) who are qualified for coverage under the provincial drug benefit plan. On the contrary, data on the prescription records for the entire population are available in this study. Finally, there was a similar work published recently with the same objective of this present study, but that paper compared two diagnosis-based comorbidity indices with two medication-based ones [40]. In contrast, this study is relatively more comprehensive and thus brings significant added value to the literature since we have evaluated four diagnosis-based comorbidity indices and two medication-based comorbidity indices as well as analyzed three target years.

Despite the strengths of this study, our findings need to be interpreted with caution with regard to some limitations which are inherent in retrospective claims data analysis. Firstly, data on identification of comorbid conditions of patients are limited to diagnoses recorded by clinicians via medical claims within the time frame studied. Additionally, while we take advantage of a population-based database in evaluating risk adjustment performance of various comorbidity measures, one limitation is that comorbid conditions have been shown to be under-ascertained in administrative claim data when compared with medical records as data sources [5–7]. Lastly, more cautiousness is needed when applying research findings of this study to patients with chronic diseases in general, given that the current results are based on COPD patients and may not be generalized to other patient groups.

In summary, this study provides complementary insight valuable to researchers intending to select adequate comorbidity indices for use with health care utilization databases. This work has ascertained that comorbidity indices are significant predictors of medical expenditures and mortality of COPD patients. Based on the study findings, we propose that when designing the payment schemes for patients with chronic diseases (e.g., COPD, the study population of this analysis), the health authority should make adjustments in accordance with the burden of health care caused by comorbid conditions. In addition, the managerial utility could be carried out by identifying high-risk patients for integrated care planning since care comorbidities as comorbidities are associated with higher healthcare resource utilization and lower health-related quality of life.

## Supporting information

**S1 Table. Results of binary logistic regression models with all covariates.**
(DOCX)

## Acknowledgments

This study is based in part on data obtained from the National Health Insurance Research Database provided by the National Health Insurance Administration of Ministry of Health

and Welfare, and managed by the National Health Research Institutes, Taiwan. No potential conflicts of interest relevant to this article were reported. The interpretation and conclusions contained herein do not represent those of the National Health Insurance Administration, the Ministry of Health and Welfare, or the National Health Research Institutes, Taiwan.

## Author Contributions

**Conceptualization:** Huei Guo Ie, Chao-Hsiun Tang, Mei-Ling Sheu, Kuo-Cherh Huang.

**Data curation:** Kuo-Cherh Huang.

**Formal analysis:** Huei Guo Ie, Hung-Yi Liu.

**Funding acquisition:** Kuo-Cherh Huang.

**Investigation:** Huei Guo Ie, Tuan-Ya Tsai, Bi-Li Chen, Kuo-Cherh Huang.

**Methodology:** Huei Guo Ie, Chao-Hsiun Tang, Mei-Ling Sheu, Ning Lu, Kuo-Cherh Huang.

**Project administration:** Huei Guo Ie, Kuo-Cherh Huang.

**Resources:** Huei Guo Ie, Hung-Yi Liu, Kuo-Cherh Huang.

**Software:** Huei Guo Ie, Hung-Yi Liu, Kuo-Cherh Huang.

**Supervision:** Kuo-Cherh Huang.

**Validation:** Huei Guo Ie, Chao-Hsiun Tang, Mei-Ling Sheu, Ning Lu, Kuo-Cherh Huang.

**Writing – original draft:** Huei Guo Ie, Kuo-Cherh Huang.

**Writing – review & editing:** Huei Guo Ie, Chao-Hsiun Tang, Mei-Ling Sheu, Hung-Yi Liu, Ning Lu, Tuan-Ya Tsai, Bi-Li Chen, Kuo-Cherh Huang.

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
