## [Decision Letter · Decision Letter 0]

29 Mar 2022

PONE-D-21-18309

Evaluation of the discriminative power of diagnosis-based and medication-based comorbidity indices in patients with chronic obstructive pulmonary disease

PLOS ONE

Dear Dr. Huang,

Thank you for submitting your manuscript to PLOS ONE. After careful consideration, we feel that it has merit but does not fully meet PLOS ONE’s publication criteria as it currently stands. Therefore, we invite you to submit a revised version of the manuscript that addresses the points raised during the review process.

We look forward to receiving your revised manuscript.

Kind regards,

Jun Hyeok Lim, M.D.

Academic Editor

PLOS ONE

Journal Requirements:

2. In your ethics statement in the Methods section and in the online submission form, please provide additional information about the data used in your retrospective study. Specifically, please ensure that you have discussed whether all data were fully anonymized before you accessed them and/or whether the IRB or ethics committee waived the requirement for informed consent. If patients provided informed written consent to have data from their medical records used in research, please include this information.

3. Thank you for stating the following financial disclosure: "This study was supported by the Ministry of Science and Technology in Taiwan (grant number MOST 105-2410-H-038-010)."

Reviewers' comments:

Reviewer's Responses to Questions

**Comments to the Author**

1. Is the manuscript technically sound, and do the data support the conclusions?

Reviewer #1: Yes

Reviewer #2: No

2. Has the statistical analysis been performed appropriately and rigorously? 

Reviewer #1: Yes

Reviewer #2: No

3. Have the authors made all data underlying the findings in their manuscript fully available?

Reviewer #1: Yes

Reviewer #2: No

4. Is the manuscript presented in an intelligible fashion and written in standard English?

Reviewer #1: Yes

Reviewer #2: Yes

5. Review Comments to the Author

Reviewer #1: The authors present a novel evaluation of model performance for predicting health care utilization in COPD based off administrative data. This is complementary to other previously-done evaluations, some of which are cited. The report is well written and clear. The methodology is logical, but leaves a few questions, which if answered, would strengthen the report.

Major comments:

- Please explain your choice of years for inclusion in more detail. These data are now ≥9 years old. Was newer data not available? Was the choice related to the ICD-9 vs -10? If the latter, there are CCI and ECI algorithms available for ICD-10 as well.

- Did you consider fitting a model with both RxRisk-V and ECI since they were the front-runners in model improvement?

- I don't see a clear comparison with statistical tests of significance of your model stability over the 3 time points. If these weren't done, I would recommend them; if they were done, please make it more clear in your reporting.

- The discussion could be more detailed about the additive value of this investigation.

Minor comments:

- A prior evaluation of CCI versus ECI for utilization was omitted from the literature review. Consider citing Buhr RG et al, BMC Health Services. PMID 31615508.

- There's a typo on page 8, line 154, where "209" is presented instead of "2009"

Reviewer #2: The manuscript compares the improvement in prediction of outcomes in COPD at adding different comorbidity indices to a baseline prediction model. Results show that overall all comorbidity indices work well. themanuscript is clearly written in introduction and discussion. However, some aspects of Methods and Results require clarification and eventually additional work.

Major comments

1. Objective. The title of the manuscript suggests that the different indices are assessed with respect to discrimination, although actually discrimination results (capacvity of the model to order individuals according to their risk of presenting the event) are not shown. This Reviewer suggests to conduct a real discrimination analysis or to change the objective (including title) according to the actual analysis.

2. Comorbidity indices. It would help the reader to understand a bit more on each of the indices. How were they selected? How many and which variables do they include? how does the scoring work? This information would help not only to understand the manuscript but mostly to take decisions based on your results.

3. Study subjects. Patients were selected based on having had an admission due to COPD. The information provided shows a specificity of 78%, which is not high. It is surprising that some patients undergo surgery during the index admission - which surgery would require a COPD admission (typically an exacerbation)? I suggest additional criteria are applied (availability of lung fucntion tests, some drugs treatment) to improve the validity of the COPD diagnosis.

4. Sample size. The number of patients having a in-hospital death is very small. Was the study powered for it? If not, I suggest removing this part of the analysis.

5. Analysis, regression models. It is not clear how a logistic model is fitted for the outcome "expenditure", which is continuous in nature. Please clarify.

6. Analysis, discrimination. I suggest to add, for each index, the predicted outcome probability according to index scores, to assess (at least visually) discrimination.

Minor comments

7. I suggest to show the actual regression models with all their covariates as supplementary information .

6. PLOS authors have the option to publish the peer review history of their article (what does this mean?). If published, this will include your full peer review and any attached files.

Reviewer #1: No

Reviewer #2: No

---

## [Author Response · Author response to Decision Letter 0]

7 May 2022

Please kindly refer to the attached "Response to Reviewers" file. Thank you.

---

## [Decision Letter · Decision Letter 1]

30 May 2022

PONE-D-21-18309R1Evaluation of risk adjustment performance of diagnosis-based and medication-based comorbidity indices in patients with chronic obstructive pulmonary diseasePLOS ONE

Dear Dr. Huang,

Thank you for submitting your manuscript to PLOS ONE. After careful consideration, we feel that it has merit but does not fully meet PLOS ONE’s publication criteria as it currently stands. Therefore, we invite you to submit a revised version of the manuscript that addresses the points raised during the review process.

We look forward to receiving your revised manuscript.

Kind regards,

Jun Hyeok Lim, M.D.

Academic Editor

PLOS ONE

Journal Requirements:

Reviewers' comments:

Reviewer's Responses to Questions

**Comments to the Author**

1. If the authors have adequately addressed your comments raised in a previous round of review and you feel that this manuscript is now acceptable for publication, you may indicate that here to bypass the “Comments to the Author” section, enter your conflict of interest statement in the “Confidential to Editor” section, and submit your "Accept" recommendation.

Reviewer #1: All comments have been addressed

Reviewer #2: (No Response)

2. Is the manuscript technically sound, and do the data support the conclusions?

Reviewer #1: (No Response)

Reviewer #2: Yes

3. Has the statistical analysis been performed appropriately and rigorously? 

Reviewer #1: (No Response)

Reviewer #2: Yes

4. Have the authors made all data underlying the findings in their manuscript fully available?

Reviewer #1: (No Response)

Reviewer #2: Yes

5. Is the manuscript presented in an intelligible fashion and written in standard English?

Reviewer #1: (No Response)

Reviewer #2: Yes

6. Review Comments to the Author

Reviewer #1: (No Response)

Reviewer #2: The authors have provided clear responses to comments raised. A couple of minor issues.

1. THe question on statistical power is not irrelevant, I understand the sample size is what it is. However, the issue with low power is that results may be simply wrong (because one can not discard chance). I strongly suggest that the authors add a sentence/short paragraph with statistical power calculations (whatever the result - 10%, 40% or 99%) and let the reader interpret.

2. The description of the diverse indices is very useful but maybe too extended for the introduction and completely cuts the flow. I suggest to keep it short in introduction and detailed in methods.

7. PLOS authors have the option to publish the peer review history of their article (what does this mean?). If published, this will include your full peer review and any attached files.

Reviewer #1: No

Reviewer #2: No

---

## [Author Response · Author response to Decision Letter 1]

7 Jun 2022

Please kindly refer to the attached Response-to-Reviewers file.

---

## [Editor Report · Decision Letter 2]

12 Jun 2022

Evaluation of risk adjustment performance of diagnosis-based and medication-based comorbidity indices in patients with chronic obstructive pulmonary disease

PONE-D-21-18309R2

Dear Dr. Huang,

We’re pleased to inform you that your manuscript has been judged scientifically suitable for publication and will be formally accepted for publication once it meets all outstanding technical requirements.

Kind regards,

Jun Hyeok Lim, M.D.

Academic Editor

PLOS ONE
---

## [Editor Report · Acceptance letter]

29 Jun 2022

PONE-D-21-18309R2 

Evaluation of risk adjustment performance of diagnosis-based and medication-based comorbidity indices in patients with chronic obstructive pulmonary disease 

Dear Dr. Huang:

I'm pleased to inform you that your manuscript has been deemed suitable for publication in PLOS ONE. Congratulations! Your manuscript is now with our production department. 

Kind regards, 

on behalf of

Dr. Jun Hyeok Lim 

Academic Editor

PLOS ONE